# Green Oxidation of Amines by a Novel Cold-Adapted Monoamine Oxidase MAO P3 from Psychrophilic Fungi *Pseudogymnoascus* sp. P3

**DOI:** 10.3390/molecules26206237

**Published:** 2021-10-15

**Authors:** Iga Jodłowska, Aleksandra Twarda-Clapa, Kamil Szymczak, Aneta M. Białkowska

**Affiliations:** 1Institute of Molecular and Industrial Biotechnology, Lodz University of Technology, 90-537 Lodz, Poland; iga.jodlowska@dokt.p.lodz.pl (I.J.); aleksandra.twarda-clapa@p.lodz.pl (A.T.-C.); 2Institute of Natural Products and Cosmetics, Lodz University of Technology, 90-537 Lodz, Poland; kamil.szymczak@dokt.p.lodz.pl

**Keywords:** psychrophilic fungi, monoamine oxidase, oxidative deamination

## Abstract

The use of monoamine oxidases (MAOs) in amine oxidation is a great example of how biocatalysis can be applied in the agricultural or pharmaceutical industry and manufacturing of fine chemicals to make a shift from traditional chemical synthesis towards more sustainable green chemistry. This article reports the screening of fourteen Antarctic fungi strains for MAO activity and the discovery of a novel psychrozyme MAOP3 isolated from the *Pseudogymnoascus* sp. P3. The activity of the native enzyme was 1350 ± 10.5 U/L towards a primary (*n*-butylamine) amine, and 1470 ± 10.6 U/L towards a secondary (6,6-dimethyl-3-azabicyclohexane) amine. MAO P3 has the potential for applications in biotransformations due to its wide substrate specificity (aliphatic and cyclic amines, pyrrolidine derivatives). The psychrozyme operates at an optimal temperature of 30 °C, retains 75% of activity at 20 °C, and is rather thermolabile, which is beneficial for a reduction in the overall costs of a bioprocess and offers a convenient way of heat inactivation. The reported biocatalyst is the first psychrophilic MAO; its unique biochemical properties, substrate specificity, and effectiveness predispose MAO P3 for use in environmentally friendly, low-emission biotransformations.

## 1. Introduction

Monoamine oxidases (MAOs) (E.C. 1.4.3.4) belong to the class of oxidoreductases and catalyze the oxidative deamination of biogenic and xenobiotic amines using flavin adenine dinucleotide (FAD) as a prosthetic group and oxygen as an ultimate acceptor of hydrogen atoms [1]. Flavin MAOs are widespread in nature since they are synthesized by bacteria as well as lower and higher eukaryotes [2,3].

All MAOs have an almost identical FAD-binding domain (Rossmann domain) in their structure and catalyze double-displacement bisubstrate reactions, in which the substrates are: an amine and molecular oxygen (Figure 1).

The reactions in which one of the products is released before the second substrate binds to the enzyme’s active site are also known as Ping Pong reactions. This means that oxidative deamination is a two-step reaction. In the first step—reductive—FAD acts as an acceptor of two hydrogen atoms that are detached from the substrate and are reduced to FADH_2_. In the second step—oxidative—hydrogen atoms from the reduced FADH_2_ are transferred by the enzyme to the oxygen molecule, which is reduced to hydrogen peroxide, and FAD returns to its oxidized form. Simultaneously, the imine intermediate, which leaves the enzyme’s active site before the binding of the oxygen molecule, is spontaneously hydrolyzed to an aldehyde or a ketone [4,5].

The best known and characterized enzymes from the MAO family are of human origin. In mammalian tissues, two distinct forms of MAO have been found, MAO A and B, which display differences in substrate and inhibitor specificity [2]. MAO A most preferably oxidizes biogenic amines, e.g., serotonin, and is inhibited by low (10^−8^ M) concentrations of clorgyline, whereas MAO B preferentially oxidizes xenobiotic amines such as β–phenylethylamine and is inhibited by low (10^−7^ M) concentrations of deprenyl [6]. Both MAO isoforms are attached to the outer mitochondrial membrane and contain a covalently attached flavin cofactor [5]. Fluctuations in the levels of these enzymes are thought to contribute to neurological diseases such as Alzheimer’s and Parkinson’s as well as behavioral diseases [7].

MAOs are also present in lower eukaryotes and bacteria where their main biological role is to provide the organism with ammonium. Microorganisms are able to synthesize MAOs while cultured on a medium containing monoamines or diamines as the sole nitrogen source [3]. The best characterized MAO from lower eukaryotes is the enzyme isolated from *Aspergillus niger* (MAO N) [8]. It is characterized by broad substrate specificity, from small amines to big neurotransmitters, so it combines the specificity of MAO A and MAO B; it is also inhibited by both clorgyline and deprenyl [8,9].

The comparison of the structures of the three above-mentioned MAOs shows a similar overall fold, with the amino acids involved in FAD binding being well conserved among MAOs (Figure 1A). However, bacterial and fungal MAOs bind the flavin cofactor noncovalently (as shown for MAO N, Figure 1B), which differentiates them from their mammalian homologues [10,11].

Due to the broad substrate specificity of fungal MAO N and its ability to catalyze deracemization of chiral amines, it was highly desirable for applications in biocatalysis. MAO N became a model for directed evolution experiments; all the variants D3–D11 were created from the wild-type enzyme by point mutations inside and around the active site [10]. These MAO N mutants can be used in regio- and stereoselective cascade reactions of chiral disubstituted pyrrolidines which constitute a significant structural motif present in various natural products and many active pharmaceutical ingredients (APIs) [12,13]. D3 (mutations of N336S, I246M, and M348K) and D5 (mutations of I246M, N338K, M348, T384N, and D385S) variants are able to catalyze deracemization of primary and cyclic secondary amines with a separation level of around 90% [14]. The D11 (mutations of I246M, N338K, M348, T384N, D385S, W430G, F210M, L213T, M242Q, and I246T) variant has great potential in the synthesis of APIs such as levocetirizine and solifenacin. Other variants, e.g., D5 and D9 (mutations of I246M, N338K, M348, T384N, D385S, F210L, L213T, M242Q, I246T, and W430H), are used for the racemic separation of natural alkaloids such as coniine and eleagnine, respectively [12]. Variant D9 was recently used for the biosynthesis of pyridine derivatives due to its aromatizing activity. Pyridine can be found in many natural compounds (vitamins and NADP/NADPH), as well as in drugs (e.g., anticancer agent sorafenib) and can be used in the flavour and fragrance industry [15]. MAO N is a great example of how we can apply biocatalysis in the pharmaceutical industry and manufacturing of fine chemicals, where the previous solution was only chemical synthesis. The use of enzymes in amine oxidation makes the process greener and brings a reduction in overall costs.

In this paper, we report the identification of a novel MAO isolated from an Antarctic strain—*Pseudogymnoascus* sp. P3 (and suggest the name MAO P3), which is the first native psychrophilic MAO able to catalyze the oxidative deamination of primary and secondary amines in low temperatures and with high yields of reactions. Using cold-adapted enzymes in biocatalysis allows for higher regio- and stereoselectivity of desired products and lower overall costs of production due to the decreased temperatures of bioconversion [16].

## 2. Results and Discussion

### 2.1. Identification of a Potential MAO Producer from Antarctic Fungi Strains

Numerous MAOs of various origins have already been identified, however, to date, none of these enzymes described in the literature so far were derived from a cold-adapted microorganism [17]. The research presented in this work was aimed at identifying the first psychrophilic MAO. Fourteen Antarctic fungi strains belonging to the collection of psychrophilic microorganisms at IMIB TUL were cultured for 14 days in an induction medium containing *n*-butylamine as the sole nitrogen source. The best concentration for MAO biosynthesis in the native host was optimized to the value of 0.10% of *n*-butylamine (Table 1); lower concentrations did not enhance the level of MAO biosynthesis, whereas the higher ones (0.2% and above) acted deadly on the psychrophilic strains, which is connected with the killing properties of amines [3].

In general, the rate of growth of fungi in the induction medium was weaker than in the growth medium with 0.1% NaNO_3_ as a nitrogen source (see the Appendix A). For strains P1, P13, P14, P18, and P19, the amount of filtrated wet biomass was lower than 100 mg in the induction medium. For the rest of the psychrophilic strains, the amount of biomass was between 200 mg (P3, P7, P5, P8, P9, P11, and P15) to almost 1 g for the P12 strain. The collected biomass of psychrophilic fungi was used to extract intracellular proteins and determine the ability to produce MAO. The preliminary screening for MAO activity was performed using a qualitative strip test for H_2_O_2_ detection (see the Appendix A). Results were then confirmed by measuring MAO activity: the initial activity of MAO was measured in the intracellular protein extract of all strains against primary (*n*-butylamine) and secondary (6,6-dimethyl-3-azabicyclohexane) amines. This experiment allowed for the quantitative identification of potential producers of MAO (Figure 2A).

Most of the strains that grew poorly in the induction medium displayed no (P8, P13, P14, P16, P18, and P19) or almost no (P1) MAO activity. Very little activity was detected in the strains P9, P11, and P12. Finally, strains P3, P5, P7, and P15 were identified as active towards the tested amine substrates.

In all cases except for P3, the activity towards a primary amine was higher than for a secondary. The highest activity towards both primary (*n*-butylamine) and secondary (6,6-dimethyl-3-azabicyclohexane) amines was observed for the P3 strain of the psychrophilic fungi, at the level of 910 ± 4.8 U/L and 959 ± 7.2 U/L, respectively. In further experiments, the inoculation method was changed and involved a preculture of the P3 strain on a solid growth medium for 7 days at 20 °C, which caused a significant increase in the activity towards primary and secondary amines for most of the already identified MAO-producing psychrophilic strains (Figure 2B).

MAO activity increased by almost 50% only from changes in the cultivation procedure. For the P3 strain, the level of activity increased to 1350 ± 10.5 U/L for a primary, and 1470 ± 10.6 U/L towards a secondary amine. The results of this screening carried out on fourteen strains of Antarctic filamentous fungi showed that these psychrophiles can be MAO producers, thus confirming the theory that in general, most microorganisms grown on a medium with primary or secondary amines as the sole nitrogen source are able to biosynthesize amine oxidases, including MAOs. This ability probably originated from their natural adaptation to grow in such conditions [3,9]. Most of the active strains showed higher activity towards *n*-butylamine; only one had higher activity towards the secondary amine (P3 strain), where both were monoamines, aliphatic and cyclic, respectively. Activity towards monoamines is typical for MAO enzymes, e.g., for native MAO N (amylamine, *k*_cat_ = 1000 min^−1^; benzylamine, *k*_cat_ = 371 min^−1^) [18]. However, the ability of the P3 strain to oxidize 6,6-dimethyl-3-azabicyclohexane is promising for the use of MAO enzymes in various biotransformations involving substituted pyrrolidine.

### 2.2. Characteristics of the P3 Strain

Strain P3, which showed the highest level of MAO activity, was genetically identified as the *Pseudogymnoascus* sp. (reclassified from *Geomyces* sp.) based on the ITS1/5.8S rRNA/ITS2 and 28S rRNA sequence analysis. The sequences were deposited in GenBank with the following accession numbers: KU687117 and KU687118 for ITS1/5.8S rRNA/ITS2 and 28S rRNA, respectively. Both fragments were more than 99% identical to ITS1/5.8S rRNA/ITS2 (accession number MT133864) and 28S rRNA (accession number MT509382) sequences of *Pseudogymnoascus* sp. (*Pseudeurotiaceae*, *Ascomycota*). The analysis of the phylogenetic trees based on ITS1/5.8S rRNA/ITS2 sequences confirmed that the P3 strain is closely related to other *Pseudogymnoascus* species, rather than to other meso- or psychrophilic fungi belonging to *Ascomycota* (see the Appendix A). Fungi that belong to this genus can be found in many ecological niches around the world, function as saprotrophs, and are psychrophilic or psychrotolerant; some species can even survive in the permafrost layer. The most recognized representative is *Pseudogymnoascus destructans*, which causes a deadly disease of bats called white-nose syndrome [19]. The morphology of *Pseudogymnoascus* sp. is incredibly varied and depends on the growth conditions [19], which was also observed for the P3 strain. This psychrophile was not able to proliferate in temperatures higher than 20 °C and had the ability to grow on media containing waste from the food industry such as apple pomace, brewing spent grain, and orange and watermelon peels (characteristics presented in the Appendix A). While growing on these media, *Pseudogymnoascus* sp. P3 was able to produce extracellular enzymes such as cellulases, pectinases, and xylanases. It could also degrade tributyrin and gelatin. Another interesting feature of the strain was its ability to grow in the presence of NaCl in a range of 1–10%, which is in agreement with the fact that most *Pseudogymnoascus* sp. are able to tolerate a wide range of salt concentrations [19].

### 2.3. Inhibition Analysis

To test the influence of an irreversible MAO inhibitor—clorgyline—on the novel MAO isolated from the *Pseudogymnoascus* sp. P3 (MAO P3), the enzyme was preincubated with different concentrations of the inhibitor, as the inactivation process is known to be time-dependent [8]. The IC50 value calculated from the inhibition experiments for MAO P3 was 1.8 × 10^−8^ M (Figure 3), whereas the IC50 value for MAO A is 1.2 × 10^−9^ M [6].

Results showed that even a small concentration of this acetylene inhibitor impeded the conversion of the amine substrate, which is explained by creating an adduct of the oxidised inhibitor with the reduced FAD [8]. It could be suggested that the novel MAO P3 may share structural similarities (especially within the Rossmann domain) with its homologues MAO A and MAO N [6], which could explain the fact of inhibition by the same compound.

### 2.4. Purification of MAO P3

After preliminary analyses, MAO P3 was purified by affinity chromatography from the dialyzed intracellular protein extract of *Pseudogymnoascus* sp. P3 to obtain a homogenous preparation without the need for further purification steps. The HiTrap Blue column is commonly used for the purification of oxidoreductases [20]. The elution profile shows that apoprotein was eluted by buffer B during the first step of the gradient (20%, Figure 4A). Purification on the HiTrap Blue column was a very effective procedure by which we were able to obtain a homogenous protein with the yield of the process being around 31% with twice higher specific activity (Figure 4C). The SDS-PAGE analysis showed only one band that had a mass around 60–65 kDa (Figure 4B), which is similar to the masses of other enzymes of this type such as the MAO N isolated from *A. niger* (55 kDa) [10].

The cold-adapted MAO P3 showed typical MAO features such as noncovalent binding of FAD [8], which is essential for the proper activity of the enzyme [10,21]. In the dialysis experiment, with the use of KBr, it was possible to dissociate the cofactor molecule from the enzyme which resulted in a complete loss of activity, which is typical for all flavoenzymes including MAO N [8], arsenate oxidase [22], and d-amino acid oxidases [23]. MAO P3 was purified as an apoenzyme that required reconstitution by the addition of FAD for full activity. The optimized procedure for reconstitution of active MAO P3 involved the incubation of the apoenzyme with 25 μM of FAD for 30 min at 20 °C. The holoenzyme’s reconstitution was confirmed by the measurement of absorption spectrum (Figure 4D), at which the shift in absorbance between free FAD (absorbance maxima for the cofactor at the wavelengths of 263 nm, 375 nm, and 450 nm) and holoenzyme (maxima at 272 nm, 357 nm, and 455 nm; the absorption maxima characteristic for the oxidized flavoprotein [21]) was observed. The suggested explanation is that the shift of the maximum from 263 nm to 272 nm was probably caused by the binding of the AMP molecule, a part of FAD; the maximum from 375 nm was lowered to 370 nm, while in the case of 450 nm, it was increased to 455 nm, probably due to the interaction between the isoalloxazine ring and the tyrosine’s (Y389 in MAO N) side chain [10,24]. The ease with which the FAD has been detached from the enzyme indicates its noncovalent binding. This proves that fungal MAO P3 binds flavins in a similar way to other fungal MAOs and suggests a high homology and structural similarity within the FAD-binding Rossmann domain [10,11]. The psychrophilic enzyme’s action was markedly accelerated by the extra FAD addition during the reaction. When 0.2 mM of FAD was included in the reaction mixture, the activity was increased two-fold compared to an active form of a holoenzyme without supplementation of FAD.

### 2.5. Temperature Profile of MAO P3

The optimal temperature of catalysis by the novel MAO P3 is 30 °C (Figure 5A). Other cold-adapted oxidoreductases exhibit similar optimal temperatures, e.g., glucose oxidase from *Cladosporium neopsychrotolerans* SL16 which had the highest activity at 20 °C [25]. On the other hand, certain enzymes isolated from psychrophilic strains exhibit high optimal temperatures, e.g., glucose oxidase isolated from the Antarctic yeast *Goffeauzyma gastric *had a T_opt_ of 64 °C [26]. Until now, no thermophilic MAO was described, however, a d-amino acid oxidase from *Rubrobacter xylanophilus* with a T_opt_ of 65 °C and stable in the range 20–60 °C was reported [23]. The unique feature of MAO P3 is its ability to maintain from 25 to 75% of maximum activity in the temperature range from 0 to 20 °C (Figure 5A). This feature predisposes the enzyme to efficient biocatalysis at temperatures lower than its mesophilic homologues, including MAO N operating at 37 °C [27].

The temperature stability profile shows that the enzyme is rather unstable above 50 °C after 30 min of incubation (Figure 5B), which characterizes the majority of cold-adapted oxidoreductases, e.g., glucose oxidase (*Cladosporium neopsychrotolerans* SL16), which was thermostable up to 50 °C for 1 h [25]. It is noticeable that by extending the time of incubation, the thermostability of MAO P3 decreases rapidly (Figure 5B). During all tested times of incubation, the enzyme showed activity above 90% of the maximum in a rather narrow temperature range—from 20 to 30 °C. These properties classify this biocatalyst to the group of enzymes adapted to catalysis at low temperatures, namely, psychrophilic enzymes, thus providing evidence for the first cold-adapted MAO.

### 2.6. pH Profile of MAO P3

The pH activity profile of MAO P3 shows that the optimal pH was at 7.0 (Figure 5C). A typical characteristic of MAO P3 and other enzymes that belong to the class of oxidoreductases and flavoproteins is their optimal pH. For a vast majority, it is around neutral (pH 7–8), e.g., glucose oxidase isolated from *Cladosporium neopsychrotolerans* SL16 exhibited maximal activity at pH 7, and thermophilic d-aspartate oxidase from *Thermomyces dupontii* at pH 8 [28], or slightly alkaline pH, as in the case of thermophilic d-amino acid oxidase isolated from *Rasamsonia emersonii* [29]. The cold-adapted MAO P3 was stable (>80%) from pH 6.5 to 8.5 (Figure 5D), which is consistent with the data on the pH stability of psychrophilic oxidases in the range of pH 6.0–10.0 [25]. MAO P3 is not active below pH 5, as flavoproteins undergo irreversible aggregation at acidic pH [20].

### 2.7. Bioconversion of Amines and MAO P3 Substrate Specificity

The ability of MAO P3 to catalyze the oxidation of various amines was evaluated using sixteen substrates differing in the number of carbons and substitutions (Table 2). The analyzed compounds ranged from simple primary aliphatic or cyclic amines to more structurally complicated primary and bicyclic secondary amines.

The psychrophilic enzyme was able to catalyze the oxidative deamination of substrates *sec*-butylamine (**1**), cyclopentylamine (**2**), pyrrolidine (**3**), α-methylbenzylamine (**5**), hexamethyleneimine (**6**), 6,6-dimethyl-3-azabicyclohexane (**9**), 3-azabicyclo[3.3.0]octane (**10**), and indoline (**12**), for which the rates of oxidation relative to compound **9** are shown in Table 3. The compounds 2-azabicyclo[2.2.1]hept-5-en-3-one (**7**) and pyridine (**8**) were used as the negative controls because according to the literature, amides (**7**) or azines (**8**) cannot be oxidized by MAOs [12]. The highest level of activity was observed for substrate **9**, the secondary amine which was also the most actively converted by MAO P3 in the spectrophotometric assay (Figure 2), whereas the lowest rate of oxidation was observed for substrate **12**. MAO P3, similarly to MAO N, was not able to convert a diamine–putrescine (**15**) or a polyamine–spermidine (**16**) [8]. The lack of activity towards **15** and **16** could be caused by the differences in the substrate structure, as the structure-to-function comparison of plant polyamine oxidase (PAO) and MAO B revealed that those enzymes differ in the mechanism of reaction and protein folding, which influences the substrate-binding sites and ultimately affects their diverse substrate specificity [30]. The compounds 1-methylpyrrolidine (**4**), cis-8-azabicyclo[4.3.0]nonane (**11**), 1-(2-aminoethyl)pyrrolidine (**13**), and benzylhydrylamine (**14**) also remained unconverted even after 24 h of incubation at 30 °C with shaking.

To evaluate the enzyme’s affinity for the substrates that MAO P3 was able to oxidize, the *V*_max_ and *K*_m_ values were determined based on the Beer-Lambert curve (Table 3, plots available in the Appendix A).

A very distinct feature of MAO P3, visible even at the stage of the initial screening, is its high activity towards a secondary amine and a pyrrolidine derivative, 6,6-dimethyl-3-azabicyclohexane (**9**), for which the enzyme also has the highest affinity (*K*_m_ = 2.4 mM). This differentiates the reported psychrozyme from its mesophilic counterpart, native MAO N, which was not able to oxidize such structurally complex substrates. Native MAO N was active towards aliphatic amines (e.g., propylamine, butylamine, hexylamine) and had the highest affinity towards 2-phenethylamine (*K*_m_ = 0.1 mM) [8]. MAO P3 is also able to oxidize at a lower rate than pyrrolidine itself (**3**, 36%), as well as another derivative, 3-azabicyclo[3.3.0]octane (**10**, 67%). Analyzing the structure of the tested pyrrolidine derivatives, it can be noticed that with the increase in the size of the substituent (**9** vs. **10**), the affinity towards the substrate decreases, which can be caused by the size of the catalytic pocket, as observed in the case of native MAO N [8]. This was not the case for recombinant MAO N D5, where the rate of initial oxidation increased together with the size of the substituent e.g., for 3-azabicyclo[3.3.0]octane and *cis*-8-azabicyclo[4.3.0]nonane it was 16% and 19%, respectively [31]. MAO P3 showed no activity towards a tertiary amine–substrate **4,** as well as towards a diamine (**13**), where a primary ethylamine is attached to the nitrogen in the pyrrolidine ring. The result obtained for compound **4** was similar to the case of MAO N D5, however variant D5 exhibited activity towards tertiary amines such as *N*-methyl-2-phenylpyrrolidine [27,32]. Novel MAO P3 displayed the ability to oxidize a secondary amine containing a ring with eight carbons (**6**), with an oxidation rate at the level of 30% and a high affinity (*K*_m_ = 2.64 mM). MAO N D5 also exhibited a high affinity towards a secondary amine (**6**), which was higher than for compound **10** as in the case of cold-adapted MAO P3.

What is surprising is the fact that MAO P3 is able to oxidize compound **12** (with a low 9% rate) with the *K*_m_ value of 20.4 mM. It could be expected that the enzyme should then be able to convert substrate **11** as well, also a pyrrolidine derivative, however not containing an aromatic ring. Such a stable moiety was blocking the conversion in the D5 variant of MAO N (mutant I246M, N338K, M348, T384N, and D385S) through steric hindrance [27]. However, the issue of the binding of the substrates **11** and **12** could only be deciphered using the methods of structural biology (e.g., by X-ray crystallography) or by molecular modelling. As mentioned before, MAO P3 is not able to oxidize the bulky substrate **14**, which probably can be explained by a steric hindrance of two phenol groups, similarly to native MAO N which was able to oxidize simple aliphatic amines (e.g., amylamine, benzylamine) and exhibited a lower activity towards benzylamine [27]. Accommodation of bulky substrates could be achieved in the process of protein engineering via directed mutagenesis, which was successfully performed for the MAO N mutant D10 (mutations of I246M, N338K, M348, T384N, D385S, and W430G) and D11C (mutations of I246M, N338K, M348, T384N, D385S, W430G, F210M, L213T, M242Q, and I246T). Changes introduced in and around the active site allowed for its expansion to 195 Å and 464 Å in the case of D10 and D11C, respectively, which were created based on variant D5 where the active site volume was 140 Å [12].

High affinity towards substrate **5** (*K*_m_ = 4.1 mM) showed that MAO P3 differs from native MAO N which was able to oxidize α-methylbenzylamine only at a low level (*k*_cat_ = 0.17 min^−1^). Only by incorporating two point mutations in MAO N (N338S, M348K), the enzyme displayed the activity for **5** with *K*_m_ = 0.4 mM [18]. On the other hand, the native MAO N showed a high affinity towards longer aliphatic (hexylamine, *K*_m_ = 0.2 mM) and some aromatic (benzylamine, *K*_m_ = 0.24 mM) amines and a lower affinity towards the shorter ones (*n*-butylmine, *K*_m_ = 0.6 mM; propylamine, *K*_m_ = 1.6 mM) [8]. MAO P3 displays different characteristics, as the affinity towards shorter amines (*sec*-butylamine, *K*_m_ = 18 mM) is lower than towards certain aromatics (methylbenzylamine, *K*_m_ = 4.1 mM). Presented data show that the novel native psychrophilic MAO P3 has the ability to oxidize various amine substrates, giving it a broad substrate specificity which could be further expanded by way of enzyme engineering.

The goal of green chemistry is to eliminate or limit the use and production of environmentally harmful substances. The novel native MAO P3 isolated from the cold-loving *Pseudogymnoascus* sp. P3 fits into these assumptions, as the enzyme has a broad substrate specificity directed mostly at bicyclic pyrrolidine derivatives. Cyclic amines are often bioactive agents, therefore they are widely identified in natural products and used as building blocks of pharmaceuticals. They serve as amino-acid derivative components, such as 3-azabicyclo[3.3.0]octane or 6,6-dimethyl-3azabicyclohexane, which are l-proline analogues found in anti-HCV drugs [33,34]. Due to their use, it is very important to carry out racemic resolution of substituted pyrrolidines as efficiently as possible, which is possible only with the application of enzymes such as MAOs [13,31,35].

The synthesis of amines, even the primary ones, is a fairly expensive process, however, the possibilities discovered over the course of several years allowed for their production from biomass, and more precisely, from carbohydrates derived from lignocelluloses [36]. Using this method, both aliphatic amines and pyrrolidines are obtained, which can then be oxidized to imine and then hydrolyzed to ketones or aldehydes. The latter are often used in the cosmetics or food industry. In the flavour and fragrance industry, cold-adapted enzymes are more useful because these substances evaporate more intensively at higher temperatures [37]. An ideal example is the oxidative deamination of cyclopentylamine catalyzed by MAO P3 (compound **2**, Table 3 and Table 4), the product of which, cyclopentanone, is used in many industries, ranging from the cosmetic industry where it is a precursor to create jasmine fragrances, through the chemical industry where it is used for the production of synthetic resins and rubber adhesives. It is also an important agent in the agricultural industry, as certain insecticides and pesticides are produced based on cyclopentanone [38,39].

The use of enzymes, especially oxidases, to catalyze oxidation reactions may allow the exclusion of organic catalysis using toxic catalysts, e.g., chromates, thereby reducing the amount of catalysts harmful to the environment. Some oxidoreductases are used in enzymatic and chemo-enzymatic cascades to produce bioactive natural products and APIs, such as ketoreductase and halohydrine dehalogenase which are used to synthesize the side chain of Atorvastatin. Cascade reactions with ω-transaminase and MAO N are used to produce optically pure derivatives of acyclic amines [13,31]. What is more, the use of a psychrophilic enzyme in chemical stereosynthesis contributes to a better racemic resolution of the mixture due to the high enantioselectivity of these enzymes, which is reflected by significantly lower activation energy (e.g., for cellulases from *Arthobacter* sp. and *Serratia marcescens*, E_A_ values were 71.6 and 78.2 kJ/mol, respectively) [40]. At the same time, the use of an enzyme from a cold-loving producer can reduce costs through lower energy consumption needed to run the process; the enzymes are easily inactivated by high temperatures that are essential in the food industry and they can be used in the biotransformation of substances that require low temperatures for reactions [16]. All of the above features make psychrozymes such as MAO P3 more applicable in industry. Finally, the novel cold-adapted enzyme exhibits high yields of reactions only in its native form, which leaves a space for further optimization during future studies on the recombinant enzyme.

## 3. Materials and Methods

### 3.1. Microorganisms and Growth Conditions

The strains of the Antarctic psychrophilic fungi were isolated in the Institute of Molecular and Industrial Biotechnology of the Technical University of Lodz (IMIB TUL) from a sample of soil collected at a hill inhabited by bryophytes and lichens located in the vicinity of the laboratory of biology at the Henryk Arctowski Polish Antarctic Station (62°10′ S, 58°28′ W) at the King George Island (South Shetlands) and named from P1 to P19. These strains were deposited in the own pure Antarctic strain collection at IMIB TUL. The Antarctic fungal strains were preserved as glycerol stocks. The characteristics of strain P3, which was found to be the most effective MAO producer, are available in the Appendix A.

The 125 mL cultures were inoculated from the glycerol stocks. The strains were cultured at 20 °C for 14 days with shaking at 100 rpm in 0.5 L flasks. The growth medium was composed of 3% (*w/v*) glucose, 0.1% (*w/v*) KH_2_PO_4_, 0.05% (*w/v*) MgSO_4_, 0.05% (*w/v*) KCl, and 0.1% (*w/v*) NaNO_3_, which was replaced by 0.1% (*v/v*) of *n*-butylamine (MAO inducer) in the induction medium. The concentration of *n*-butylamine was optimized (tested concentrations: 0.01, 0.03, 0.05, 0.10, 0.15, 0.20, and 0.50%) to maximize the biosynthesis of MAO. The mycelium was collected by filtration, wet mass was weighed and used for further experiments.

In the second variant of inoculation, all 14 fungal strains were first grown on solid growth medium which was composed of 3% (*w/v*) glucose, 0.1% (*w/v*) KH_2_PO_4_, 0.05% (*w/v*) MgSO_4_, 0.05% (*w/v*) KCl, 0.1% (*w/v*) NaNO_3_, and 2% agar. Solid culture inoculated from glycerol stocks was carried out for 7 days at 20 °C. Then the mycelium was washed with 5 mL of the induction medium from which 125 μL was used to inoculate the 125 mL culture in the induction medium.

### 3.2. Genetic Identification of Antarctic Fungal Strain P3

Identification of the Antarctic fungal strain P3 was based on the sequencing of the Internal ribosomal spacer 1 (ITS1)/5.8S rRNA/Internal ribosomal spacer 2 (ITS2) and 28S rRNA. Genomic DNA was isolated according to the phenol/chloroform method described by Sambrook (Sambrook and Russell, 2006). The fragments were amplified by PCR using the primers RLR3R (forward) and V9 (reverse) to isolate the sequence containing ITS1/5.8S rRNA/ITS2, and ITS5 (forward) and LR6 (reverse) to isolate 28S rRNA; primer sequences are shown in Table 4.

The PCR program consisted of a denaturation step at 94 °C for 2.15 min followed by 35 cycles of 94 °C for 1.15 min, 47.5 °C for 30 s, and 72 °C for 1 min. The final extension step was 72 °C for 7 min. PCR products were analyzed by electrophoresis in 1% agarose gel, purified using the DNA Basic Purification Kit (EURx, Gdańsk, Poland), and sequenced with the same primers as used in the PCR reaction. Results of sequencing were used for the analysis in tool BLASTn at NCBI [41].

Phylogenetic trees were generated in MEGA X software (with multiple sequence alignments performed with the MUSCLE tool at EBI as described in the Appendix A.

### 3.3. Protein Extraction

The intracellular proteins were extracted by a two-step procedure; the first step included grinding fungal mycelium with liquid nitrogen. The ground mycelium was transferred to a Falcon tube and resuspended in the extraction buffer (10 mM potassium phosphate buffer pH 7.2, 1 mM PMSF, 1 mM EDTA, 1 mM iodoacetic acid, 1% Triton X-100). In the second step of extraction, the resuspended mycelium was disrupted with glass beads for 30 s at 3000 rpm. The disruption cycle was repeated 8 times. The glass beads and cell debris were centrifuged at 5000 rpm for 15 min at 4 °C. The protein concentration was measured by the Bradford assay (Bradford reagent from Bio-Rad, Hercules, CA, USA) on a 96-well transparent plate using the MultiScanGo plate reader (ThermoFisher Scientific, Waltham, MA, USA) [42]. Proteins were stored at −20 °C until further analysis.

### 3.4. Assay of MAO P3 Activity

The activity of MAO P3 was measured by a colorimetric method which is based on the formation of orange-brownish tetraguaiacol in the presence of H_2_O_2_ and peroxidase (insert in Figure 2A). The reaction mixture contained 0.5 mL of the enzyme sample, 0.25 mL of 10 mM potassium phosphate buffer pH 7.2, 0.25 mL of 20 mM amine substrate (to identify a potential MAO producer, primary–*n*-butylamine, and secondary–6,6-dimethyl-3-azabicyclohexane, amines were used), 0.02 mL of peroxidase (90 U/mL), and 0.4 mL of 0.02 M guaiacol (all reagents from MerckMillipore, Burlington, MA, USA). The reaction mixture was incubated at 30 °C for 30 min.

The absorbance of the samples was measured at 436 nm on the UV/visible spectrophotometer (UV-1800 Shimatdsu, Kyoto, Japan). A unit of enzyme activity was defined as the amount of enzyme that catalyzes the formation of 1.0 μmol of H_2_O_2_ per min.

The method of preliminary screening for MAO activity using a strip test for H_2_O_2_ detection is described in the Appendix A.

### 3.5. Inhibition Analysis of MAO P3

The enzyme was preincubated with different concentrations (from 0.1 to 10^−8^ mM) of clorgyline (MerckMillipore, Burlington, MA, USA) for 30 min at 30 °C. MAO P3 activity was measured according to the above-mentioned procedure. Clorgyline solution (1 mM) was prepared in 10 mM potassium phosphate buffer pH 7.2.

### 3.6. Protein Purification

#### 3.6.1. FAD Removal

The dialysis experiment aiming at FAD removal prior to affinity chromatography was based on the modified method described by Zlateva et al. [21]. The intracellular protein extract was first dialyzed in a membrane with an MWCO of 10 kDa (Promega, Madison, WI, USA) for 24 h against 1 L of 10 mM potassium phosphate buffer pH 7.2 containing 1 M KBr. The buffer was changed four times and then the protein was dialyzed again for 24 h against 1 L of 10 mM potassium phosphate buffer pH 7.2, with three changes of the buffer. All the experiments were carried out at 4 °C with continuous stirring.

#### 3.6.2. Affinity Chromatography

MAO P3 was purified by affinity chromatography on a 1 mL HiTrap Blue column (GE Healthcare, Chicago, IL, USA). The column is packed with Blue Sepharose, which is used to purify proteins containing nucleotide-like cofactors, e.g., NAD, NADPH, or FAD. The column was equilibrated with buffer A (10 mM potassium phosphate buffer pH 7.2). The protein extract after dialysis was applied on the HiTrap Blue column connected to the ÄKTA purifier chromatography system (Cytiva, Marlborough, MA, USA) and eluted with 20 mL of step gradient of buffer B (1 M KCl in 10 mM potassium phosphate buffer pH 7.2). The quality of purified enzyme was checked by polyacrylamide gel electrophoresis in the presence of SDS (SDS-PAGE) according to Laemmli.

#### 3.6.3. Reconstitution of Flavoprotein

Reconstruction of flavoprotein MAO P3 was carried out in 10 mM potassium phosphate buffer pH 7.2 in a total volume of 5 mL, with different conditions of: FAD concentration (5, 10, 15, 20, 25, and 50 μM), temperature (5, 10, 15, 20, 25, and 30 °C), and time of incubation (from 10 min to 60 min). After incubation, extra FAD was removed by centrifugation at 5000 rpm, 4 °C, 20 min in Centricon concentrators of MWCO of 10 kDa (MerckMillipore, Burlington, MA, USA). The absorption spectrum was measured on a flat-bottom 96-well plate on a MultiScanGo plate reader (ThermoFisher Scientific, Waltham, MA, USA).

### 3.7. Effect of Temperature

Thermostability of MAO P3 was determined based on the measurement of the enzyme’s activity (paragraph 2.4), which was preceded by the incubation of the enzyme in temperatures from 20 to 100 °C for 30, 45, and 60 min. The influence of temperature on MAO P3 was examined by performing an enzymatic assay in various temperatures from 0 to 80 °C.

### 3.8. Effect of pH

The influence of pH on the enzyme’s stability was examined in 10 mM Britton-Robinson buffer with a pH range from 5.0 to 9.0. Before measuring MAO P3 activity (paragraph 2.4), the enzyme was preincubated for 1 h at 10 °C at different pHs. The effect of pH on the enzyme’s activity was defined by measuring the level of MAO P3 activity using 10 mM Britton-Robinson buffer with different pH from 5.0 to 9.0.

### 3.9. Biotransformations of Amine Substrates by MAO P3

Substrate specificity of MAO P3 was examined towards sixteen amine substrates (listed in Table 3) that differed in the number of carbons and various substitutions. Bioconversion was carried out in 5 mL glass tubes (Schott, Germany), where the total volume of the reaction was 2.5 mL and was comprised of 1 mL of MAO P3 (1 U/mL), where the cell-free extracts were used for the initial oxidation screening and the purified enzyme, for substrate specificity. An amount of 0.75 mL of a substrate (in the range of concentrations between 10 and 100 mM) was resuspended in 0.1 M potassium phosphate pH 7.2, and 0.75 mL of 10 mM potassium phosphate buffer pH 7.2. Initial oxidation screening was carried out for 4 h at 30 °C, 810 rpm. The substrate specificity was measured at 30 °C; reactions were carried out for 5 h, 810 rpm. In order to analyze the products of the bioconversion, samples were first neutralized by the addition of 0.25 mL of 1 M NaOH. An amount of 0.5 mL of dichloromethane was added to samples and incubated at room temperature with shaking for 1 min. The samples were centrifuged for 5 min, 5000 rpm, 4 °C. The organic phase was collected for the gas chromatography-mass spectrometry (GC-MS) analysis. The *K*_m_ values were calculated based on Beer-Lambert curves (see the Appendix A).

### 3.10. Analytical Method—GC-MS

GC-MS analyses were carried out on a Pegasus 4D (LECO, USA) apparatus, equipped with an Agilent 6890N gas chromatograph coupled with a time-of-flight mass spectrometer (LECO, Saint Joseph, MI, USA). Samples were injected with the GerstelMultiPurpose sampler (MPS 2). The GC was fitted with a BPX5 capillary column of 30 m × 0.25 mm × 0.25 µm (SGE). Helium was used as a carrier gas with a flow rate of 1.5 mL/min. The injector temperature was held at 250 °C. The oven temperature was initially held at 40 °C for 2 min, ramped up to 200 °C by 10 °C/min; the total run time was 20 min. The detector was operated in EI mode with the transfer line maintained at 280 °C. All substrates were purchased from MerckMillipore, USA, and quantified by GC analysis based on their peak areas and calibration curves prepared for each individual. GC-MS spectra are presented in the Appendix A.

### 3.11. Statistical Analysis

Calculations were performed using Microsoft Office Excel version 2007. Experimental values were reported as the means ± s.e. All calculations of statistical significance were made using GraphPad Prism5 and SPSS ver.11. Graphs were plotted using GraphPad Prism5.

## 4. Conclusions

In conclusion, this article reported the screening of fourteen Antarctic fungi strains for MAO activity, which brought the discovery of a novel cold-adapted psychrozyme MAO P3 isolated from a strain, P3, genetically identified as *Pseudogymnoascus* sp. The activity of the native enzyme at the optimal temperature of 30 °C was 1350 ± 10.5 U/L towards a primary and 1470 ± 10.6 U/L towards a secondary amine. MAO P3 has the potential for applications in biotransformations as it displayed a wide substrate specificity. Further research will include the identification of an MAO P3 gene and its recombinant expression in order to obtain enzymatic preparation with enhanced kinetic properties.

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
