# Peer review of "Green Oxidation of Amines by a Novel Cold-Adapted Monoamine Oxidase MAO P3 from Psychrophilic Fungi Pseudogymnoascus sp. P3"

_molecules, 2021, doi:10.3390/molecules26206237_

Round 1

Reviewer 1 Report

This work describes the screening of fourteen antarctic fungi strains for monoamine oxidase activity. Also the most promising enzymes from the strains are characterized to a certain degree and the psychrozyme MAO from isolated from P3 is a promising environmently friendly alternative for synthesis processes. The work is sound and well performed.

There are a few minor points that should be improved:

  • Figure 1A has to be redone. Maybe color each enzyme in a different color, or do not show only the overlay but the structures next to each other. The way it is right now is useless as one cannot see anything. In the Figure legend the Unit (Angstroms, I suppose) to the RMSD values is missing.
  • In the SI the style has to be checked again, especially if there is a space between the value (which should be) and the unit or not.
  • constants like KM and kcat should always be written in italic.
  • The highest affinities for the MAO P3 (Table 3) are still in the millimolar range, which is significantly higher than for example the affinities of native MAO N towards some substrate (reference 8). This is explained by the increase of size of the substituent which results in the decrease of affinity. This can be caused by the size of the catalytic pocket in native MAO N. It would be helpful to calculate a homology model of the P3 enzyme, or maybe the structure is already available in the EMBL-EBI alphafold databank. Then an impression about the catalytic pocket could be obtained.

Reviewer 2 Report

Paper by Bialkowska describes the discovery of a novel monoamine oxidase, so-called MAO P3, and its activity in primary and secondary amine oxidation to aldehydes/ketones.

The article is concisely written, pleasant to read and thus probably deserves publication in Molecules.

Anyway (from a synthetic chemist's point of view) some questions deserve comments:

  • The activity is expressed as a 'rate of oxidation' relative to the best substrate (n°9). I'm not familiar with this way of quantification but if this  procedure is supposed to be applied as a synthetic tool, a test let's say on a 1g scale with conversion, yield etc would have been appreciated.
  • Compound shown as n°12 is not indoline but isoindoline: Please correct the name or the structure of the compounds.
  • with secondary amines like compound 9, what is the final product ? Does the reaction stops at the aminal or a second oxidation occurs to give the dialdehyde? What is thus the faster reaction ? MAO P3 activity is qualitatively expressed by H2O2 production: Thus should the rate be divided by 2 compared to primary amines?
  • equation on fig 2 is not equilibrated: please remove the ammonium salt
  • compound 5 is chiral, racemic: any resolution observed ?
  • page 14, line 392: going from amine to amine is not a reduction ! the sentence (to be rephrased) should be 'can then be oxidized to imine and then hydrolyzed to ketones/aldehyses'
  • page 2, line 45: 'step-oxidative-hydrogen' should be corrected.
  • 'Green' in the title appears to be superfluous
  • paragraph from line 388 to 401 is meaningless and should be suppressed: Who is going to use the oxidation of an expensive cyclopentylamine to industrially produce readily available cyclopentatone ?

Round 2

Reviewer 2 Report

Most of my comments have been taken into account by the authors.

Publication is recommended.